# Accessibility and Mechanical Stability of Nanoporous Zinc Oxide and Aluminum Oxide Coatings Synthesized via Infiltration of Polymer Templates

**DOI:** 10.3390/polym15204088

**Published:** 2023-10-14

**Authors:** Khalil D. Omotosho, Zachary Lyon, Elena V. Shevchenko, Diana Berman

**Affiliations:** 1Materials Science and Engineering Department, University of North Texas, 1155 Union Circle, Denton, TX 76203, USAzacharylyon@my.unt.edu (Z.L.); 2Center for Nanoscale Materials, Argonne National Laboratory, Argonne, IL 60439, USA; 3Department of Chemistry and James Franck Institute, University of Chicago, Chicago, IL 60637, USA

**Keywords:** porous metal oxides, alumina coatings, zinc oxide coatings, porosity accessibility, mechanical properties

## Abstract

The conformal nanoporous inorganic coatings with accessible pores that are stable under applied thermal and mechanical stresses represent an important class of materials used in the design of sensors, optical coatings, and biomedical systems. Here, we synthesize porous AlO_x_ and ZnO coatings by the sequential infiltration synthesis (SIS) of two types of polymers that enable the design of porous conformal coatings—polymers of intrinsic microporosity (PIM) and block co-polymer (BCP) templates. Using quartz crystal microbalance (QCM), we show that alumina precursors infiltrate both polymer templates four times more efficiently than zinc oxide precursors. Using the quartz crystal microbalance (QCM) technique, we provide a comprehensive study on the room temperature accessibility to water and ethanol of pores in block copolymers (BCPs) and porous polymer templates using polystyrene-block-poly-4-vinyl pyridine (PS75-b-P4VP25) and polymers of intrinsic microporosity (PIM-1), polymer templates modified by swelling, and porous inorganic coatings such as AlO_x_ and ZnO synthesized by SIS using such templates. Importantly, we demonstrate that no structural damage occurs in inorganic nanoporous AlO_x_ and ZnO coatings synthesized via infiltration of the polymer templates during the water freezing/melting cycling tests, suggesting excellent mechanical stability of the coatings, even though the hardness of the inorganic nanoporous coating is affected by the polymer and precursor selections. We show that the hardness of the coatings is further improved by their annealing at 900 °C for 1 h, though for all the cases except ZnO obtained using the BCP template, this annealing has a negligible effect on the porosity of the material, as is confirmed by the consistency in the optical characteristics. These findings unravel new potential for the materials being used across various environment and temperature conditions.

## 1. Introduction

Porous polymer-inorganic hybrid and all-inorganic structures are utilized for a broad range of applications [1]. In particular, polymer-inorganic hybrid structures demonstrate excellent performance for organic solvent separations [2] and water purification [3], while porous conformal all-inorganic coatings are of great interest in the design of optical coatings [4] and sensing layers [5,6].

Polymer templates serve as an excellent platform to prepare highly porous hybrid and all-inorganic structures, opening new routes to creating porous materials for various applications, including sensors [7,8], separation membranes [2], anti-reflective coatings [9,10], and oil absorbing foams [11]. Infiltration of the polymer assemblies with different patterns and geometries with inorganic precursors is of great interest for the design of porous nanostructures with unique mechanical, electrical, magnetic, thermal, catalytic, and optical properties [10,12,13].

Two types of polymers are currently used as templates for the synthesis of polymer-inorganic hybrids and all-inorganic nanostructured materials such as block copolymers (BCPs) and porous polymers (e.g., polymers of intrinsic microporosity or porous organic frameworks) [14,15]. The sequential infiltration synthesis (SIS) process is one of the major approaches for the infiltration of polymer templates [2,7,16,17,18,19,20] with inorganic precursors. The SIS is a diffusion-controlled chemisorption of the molecules of the inorganic precursor inside the polymer template [9]. The SIS procedure followed by the removal of the polymer template via thermal annealing, UV-ozone, or oxygen plasma treatment [21,22] is widely used for the fabrication of porous all-inorganic functional structures [23]. Swelling of the polymer template in ethanol enables improvements in the diffusion of inorganic precursors [16,24] as a result of creating additional porosity in the BCP templates via a micelle opening [21] or via removal of a shorter polymer chain in the case of polymers of intrinsic microporosity (e.g., PIM-1) [10], resulting in the fabrication of the thicker coatings [21]. In the case of BCP templates, the porosity of the final all-inorganic coating can be efficiently tuned by the volume fraction of the polar domains in the BCP and by the number of SIS cycles [9,16]. In turn, when porous polymers are used as templates, the porosity of the final coatings depends on the structure of the polymer [8]. Porous AlO_x_ and ZnO-based structures are among the most promising materials for optical and sensing applications [1,7].

Optical coatings, gas sensors, as well as solvent separation or extraction materials, are exposed to various environmental conditions such as temperature fluctuations, humidity, vibrations, and physical impacts. Mechanical robustness is a critical parameter that determines the ability of the material to withstand these conditions without compromising its properties.

Here, we provide a comprehensive study on the room temperature accessibility to water and ethanol of pores in BCP and porous polymer templates using polystyrene-block-poly-4-vinyl pyridine (PS75-b-P4VP25) and polymers of intrinsic microporosity (PIM-1), polymer templates modified by swelling, and porous inorganic coatings such as AlO_x_ and ZnO synthesized by SIS using such templates. We present a comparative analysis of the penetration of water and ethanol in pores using real-time Quartz Crystal Microbalance measurements. We also analyze the mechanical stability of the as-deposited polymer templates and corresponding structures modified via solvent swelling, infiltration of the inorganic precursors, such as trimethyl aluminum and diethylzinc, and final porous AlO_x_ and ZnO conformal coatings. We also evaluate the effect of freezing and annealing on porosity and mechanical properties. This research is crucial for expanding the applicability of highly porous conformal coatings in various application systems.

## 2. Materials and Methods

Sample Synthesis: We used two types of polymer templates–a polymer of intrinsic microporosity (PIM-1), also known as 2,3,5,6-Tetrafluorophthalonitrile-3,3,3′,3′-tetramethyl-1,1′-spirobisin dane-5,5′,6,6′-tetrol co-polymer, and a block copolymer polystyrene-block-poly-4-vinyl pyridine (PS75-b-P4VP25–with molecular weights of 75,000, for the nonpolar domain (PS), and 25,000, for the polar domain (P4VP). The chemical structures of the polymers are shown in (Figure 1). PIM-1 is composed of fused rings, which makes them rigid, with sites of contortion that are provided by spiro-centers, thereby creating molecular-sized interconnected voids. The block copolymer is synthesized via anionic polymerization of the polystyrene block and the 4-vinylpyridine block. This amphiphilic diblock copolymers are able to separate into microphases. PIM-1 was purchased from Sigma Aldrich and dissolved in chloroform to prepare a polymer solution of 20 mg/mL concentration. The BCP was purchased from Polymer Source Inc. (Dorval, QC, Canada) and dissolved in toluene to make a solution of 20 mg/mL concentration. Before spin coating, the polymers were filtered through 0.4 µm pore size poly(tetrafluoroethylene) syringe filters (Fisher Scientific, Waltham, MA, USA) to remove agglomerated clusters. The polymers were spin-coated at 600 rpm and 1000 rpm on the surfaces of AT-cut (oscillating in a shear mode) QCMs and ultrasonically cleaned silicon wafers, respectively, for 50 s, followed by baking at 70 °C for 30 min to improve adhesion of the polymer films to the substrate. The mass of the spin-coated polymer films extracted from the QCM analysis was 47 ± 2 μg/cm^2^ leading to the resulting thickness of 470 ± 15 nm and 420 ± 12 nm for the BCP and PIM-1 polymers, respectively. The polymers were swelled by immersing the samples in a beaker containing ethanol on a hot plate at a temperature of 75 °C for 1 h. All the swollen polymer samples were dried at room temperature in a fume hood for 3 h to remove ethanol molecules incorporated in the polymers.

Sequential infiltration synthesis (SIS) of the polymers with AlO_x_ and ZnO was performed in a Veeco Savannah S100 ALD system. For the SIS process, the samples were placed in the reactor chamber at 90 °C, and the chamber was pumped down to a base pressure of 365 mTorr. Then, 20 sccm of nitrogen was introduced into the reactor chamber as the carrier gas prior to infiltration. The precursors used for alumina and zinc oxide films were trimethylaluminum (TMA) and diethyl zinc (DEZ), respectively. The samples were exposed to 10 cycles of TMA for AlO_x_ and 40 cycles of DEZ for ZnO, pulsing at 5 s for each cycle, with a total exposure time of 500 s, after which the excess reactant was evacuated for 600 s. Then, 20 sccm of nitrogen was further introduced into the chamber for infiltration of H_2_O, pulsing at 10 s for each cycle, with a total exposure time of 300 s, followed by purging with nitrogen (100 sccm) for 300 s to remove not infiltrated byproducts.

After infiltration of the polymers, the polymer templates were removed by the traditional oxidative thermal annealing technique in a Thermo Scientific Lindberg Blue M Furnace for 4 h at 450 °C under air flow. In the samples prepared for QCM studies, the polymer templates were removed in an UV ozone cleaner (UVOCST16x16 OES, 254 nm UV wavelength) at room temperature (RT) since the QCM crystals are incompatible with high-temperature processing. The resulting thickness of the films varied in the range of 170 ± 15 nm for AlO_x_ (PIM), 137 ± 14 nm for ZnO (PIM), 725 ± 23 nm for AlO_x_ (BCP), and 274 ± 18 nm for ZnO (BCP), respectively.

QCM Analysis: The QCM is a sensitive technique for nondisruptive in-situ quantitative analysis of the porosity evolution events [25,26]. The gold-coated AT-cut QCM crystals (1 inch in diameter) with a base resonant frequency of 5 MHz were purchased from Inficon. The resonant frequency was monitored with the aid of a QCM controller purchased from Stanford Research Systems (SRS QCM200).

In the classic approach, the change in frequency of the QCM, Δ*f*, under applied load, caused by the deposited mass, *m*, is [27,28]:(1)$$\Delta f=-\frac{{2f}^{2}}{A\sqrt{\rho_{q}{µ}_{q}}}\Delta m$$

When the QCM is immersed in a viscous liquid, the resulting change in frequency is [29]:(2)$$\Delta f=-{f_{o}}^{3/2}\sqrt{\frac{\rho_{L}\eta_{L}}{\pi {µ}_{q}\rho_{q}}}$$
where *f*_o_ is the fundamental frequency of the QCM, $\rho_{q}$ is the density of quartz (2.648 g/cm^3^), ${µ}_{q}$ is the shear modulus of quartz (2.947 × 10^11^ g cm^−1^ S^−2^), A is the QCM surface area, $\rho_{L}$ and $\eta_{L}$ are the density and viscosity of the liquid, respectively. When the QCM is immersed in water, $\rho_{L}$ = 0.9982 g/cm^3^ and $\eta_{L}$ = 0.01 g/(cm s). In the case of ethanol immersion, $\rho_{L}$ = 0.789 g/cm^3^ and $\eta_{L}$ = 0.007 g/(cm s). The expected frequency shift when the QCM is immersed in water is ~700 Hz, while the expected frequency shift when immersed in ethanol is ~500 Hz. For water penetration inside the pores, the resonant frequency change is modified by the added mass of water entrapped inside the pores [10]:(3)$$∆f=-{f_{o}}^{3/2}\sqrt{\frac{\rho_{L}\eta_{L}}{\pi {µ}_{q}\rho_{q}}}-\frac{{2f}^{2}}{A\sqrt{\rho_{q}{µ}_{q}}}∆m$$

The volume of porosity inside the formed structures can be calculated from the mass change, as a result of the additional frequency change observed, when the porous structure is immersed in water and ethanol environments. The QCM measurements in water and ethanol were used for the quantitative monitoring of the solvent penetration inside porosity and evaporation during drying.

Characterization: The JEOL JSM-7500F microscope was utilized for the scanning electron microscopy (SEM) imaging of the coatings. The thickness of the resulting nanoporous metal oxide films was measured at the cross sections using the SEM. The surface topography of the samples was performed with the aid of Bruker Multimode Atomic Force Microscopy (AFM). The images were acquired in tapping mode in the air with a scan rate of 0.5 Hz using antimony-doped silicon tips with a spring constant of 42 N/m. Contact angle measurements were conducted by the Sessile water drop (10 µL) method using a Ramé-Hart 250 contact angle goniometer. J.A. Woollam horizontal M-2000 ellipsometry system was used for the refractive index measurement and analysis of porosity. The hardness of the coatings was measured with the aid of a KLA iNano nanoindenter equipped with a Berkovich tip. The samples were mounted on the substrates using thermal glue to ensure mobilization of the samples during the measurements. The instrument was calibrated with fused silica as a reference material. In order to distinguish the effect of the silicon substrate from the hardness of our coating, we measured the hardness of the silicon wafer to be ~9.6 GPa.

## 3. Results and Discussion

Our study aims to understand (i) the effect of the templates on the infiltration efficiency with the inorganic precursors, (ii) the effect of the swelling of the PS75-b-P4VP25 and PIM-1 polymer coatings on their porosity and accessibility of their pores, (iii) the effect of the infiltration of the swollen PS75-b-P4VP25 and PIM-1 polymer coatings after SIS using TMA and DEZ on their porosity and the accessibility of their pores, and (iv) the effect of composition in all-inorganic AlO_x_ and ZnO coatings obtained via removal of the polymer templates by ozone on their porosity and the accessibility of their pores as well as their mechanical properties and ability to withstand high temperature and water freezing in their pores. Table 1 provides a detailed description of the samples and their annotations used in the figures.

Analysis of the QCM results (Figure 2) shows that the mass of the polymers (BCP and PIM) remains nearly the same after swelling in ethanol, indicating the stability of the swollen polymers. To achieve a similar mass of materials infiltrated in the swollen polymers as for AlO_x_ (TMA and H_2_O), four times as many cycles of ZnO SIS (DEZ and H_2_O) was required for ZnO infiltration. The resulting mass of porous AlO_x_ and ZnO coatings after UV ozone removal of PIM-1 are 55 µg/cm^2^ and 45 µg/cm^2^, respectively (Figure 2a), while the mass of porous AlO_x_ and ZnO coatings synthesized using the BCP template are 58 µg/cm^2^ and 39 µg/cm^2^, respectively (Figure 2b). Our data indicate that TMA efficiently infiltrates both PS75-b-P4VP25 and PIM-1 even though the polymer template has a different mechanism of interaction with TMA. Thus, prior FTIR studies indicated that metal oxide infiltration of PIM-1 occurs without the formation of chemical bonds with functional groups of PIM-1 polymers by nucleation of semi-permanent metal-organic adducts of metal oxide precursors with PIM-1′s C≡N groups [2,10]. In the case of the PS75-b-P4VP25 BCP template, the polar groups were reported to be the major sites for AlO_x_ infiltration [21,22]. The lower mass of all inorganic ZnO coatings despite four times as many SIS cycles indicates less efficient infiltration of both polymer templates with DEZ (Figure 2a,b).

To estimate the accessibility of the pores, we investigated the viscosity effect of ethanol and water on the Au/Ti AT-cut QCM crystals with the deposited corresponding coatings by monitoring the change in their frequency upon immersion in ethanol and water at room temperature (Figure 3). The mass change, ∆m in µg/cm^2^, which is indicative of the mass of solvent penetration in the pores, was evaluated from the change in frequency, -∆f (Hz). In addition, we also monitored the frequency and time it takes for the solvents to be removed from the pores during the evaporation from the coating surface at room temperature (Figure 3e,f). During evaporation, the frequency increases until it stabilizes, indicating the drying of the deposited coatings.

The absolute frequency change, ∆f, observed for bare QCM crystals is ~900 Hz, due to the viscosity effect of water and ethanol on the crystal. Therefore, any additional change in the frequency above 900 Hz is attributed to the added mass of the solvent trapped in the pores (as represented by the marks on the right axis of the graphs). For the BCP polymer immersed in water and ethanol (Figure 3a,b), we observed a negligible increase in ∆m in comparison to ∆m of the bare QCM crystal. This confirms the absence of porosity for the initial, non-swollen polymer. After swelling of the BCP in ethanol, the ∆m for the swollen BCP in water and ethanol is approximately 98 µg/cm^2^ and 80 µg/cm^2^, respectively, which is an indication of pores available for water and ethanol molecules. Such pores are formed via the opening of the micelles in the BCP. Infiltration of the swollen BCP with AlO_x_ and ZnO led to blockage of the pores and suppression of viscous behavior of the polymer, evidenced by a reduction in ∆m upon immersion in both water and ethanol. Interestingly, some pores in the AlO_x_ infiltrated BCP coating were still accessible for penetration to solvents. After removal of the BCP template in the UV ozone cleaner, the pores in ZnO coating (Figure 3c,d) were barely accessible to both water and ethanol as compared to the highly accessible porous AlO_x_ coating (Figure 3a,b) Thus, the mass of absorption of water and ethanol for the porous AlO_x_ coating is almost four times that of the porous ZnO upon their exposure to corresponding solvents. This contrasting observation could be attributed to the reduced amount of porosity of ZnO as a result of the difference in the size of the pores, their wettability, and the density of the coating. The ellipsometry analysis of the porosity in these coatings reveals that the AlO_x_ (BCP) coatings have a porosity of ~80%, while ~74% porosity was observed for the ZnO (BCP) coating. The graphs of the changes in the frequency as a function of time upon removal of the QCMs from the solvents (Figure 3e,f) suggest that ethanol expectedly evaporates faster than water due to its high vapor pressure. The time needed for the complete recovery of the initial frequency values varies for AlO_x_ and ZnO, which further supports the difference in the accessibility of the pores in all-inorganic AlO_x_ and ZnO though both structures were fabricated using the same PS75-b-P4VP25 BCP template.

In the case of PIM-1 (Figure 4), we also observe very interesting trends in the interactions of polymers and their modifications obtained via swelling and infiltration with inorganic precursors with liquid media. For example, PIM-1 demonstrates the minimal water penetration that we assume is partly a result of the size of the pores, which are reported to be less than 2 nm [30], and the hydrophobic nature of PIM-1 when interacting with water. However, we register a large change in mass for PIM-1 and swollen PIM-1 in ethanol due to the absorption of ethanol in the polymer resulting from the viscous nature of the polymer film in ethanol. SIS of the swollen PIM with ZnO precursors leads to a large increase in mass, whereas a negligible increase in mass was exhibited by the swollen PIM infiltrated with AlO_x_ precursors when immersed in water. A possible explanation for this is the nature of the surface of the infiltrated polymers. Similar to swollen BCP templates (Figure 3), the viscous nature and further swelling of the swollen PIM in ethanol were minimized by infiltration with AlO_x_ and ZnO. After the removal of the PIM polymer template, both the resulting porous AlO_x_ and ZnO coatings are accessible to water and ethanol; however, pores in ZnO obtained using the PIM-1 template seem to be more available for the ethanol. In contrast to the accessibility observations from the BCP-templated porous coatings, the porous ZnO (PIM) coating with a porosity of ~40% is more porous than the AlO_x_ (PIM) coating with ~20% porosity. As in the case of AlO_x_ and ZnO coatings obtained using the PS75-b-P4VP25 BCP template, the evaporation of ethanol occurs faster than water.

Table 2 below summarizes the effective volume of pores in the non-swollen and swollen polymers filled with water and ethanol at room temperature. It is evident that after swelling in ethanol, the BCP (PS75-P4VP25) becomes accessible to water and ethanol. In comparison to the swollen PIM-1, the volume of pores filled with water and ethanol in the swollen BCP is higher. In Table 3, we observe that the SIS with AlO_x_ and ZnO led to a reduction in pore volume accessibility to solvents in the polymers. After polymer removal, the pores become accessible to water and ethanol.

We further analyze the surfaces of the nanoporous coatings using water contact measurements and SEM imaging (Figure 5). A summary of the wetting properties shown in Figure 5a indicates that swelling of the polymers (BCP and PIM) decreases the contact angle. We observe a significant contact angle reduction for swollen BCP, which may be due to the opening of the micelles and additional porosity introduction, as well as the increase in polar groups, after swelling. This observation is in agreement with the QCM data (Figure 3a) that revealed a substantial mass change in both polymers after swelling upon their immersion into the water. On the other hand, the swollen PIM coating is less hydrophilic. In comparison to the porous AlO_x_ and ZnO coatings obtained using PIM-1, the porous AlO_x_ and ZnO coatings obtained using the PS75-b-P4VP25 BCP template are very hydrophilic, which is in agreement with the QCM data shown in Figure 3a,c, indicating the substantial mass increase upon their exposure to water. Moreover, these coatings are highly porous as evidenced by the porosity estimations and the SEM images (Figure 5b–e). The SEM images for the BCP templated coatings (Figure 5b,c) reveal a network of interconnected porous metal oxide tubes with varying size distribution. In turn, AlO_x_ and ZnO porous coatings obtained using the PIM-1 template (Figure 5d,e) are denser and smoother with the presence of spherical features.

Analysis of the mechanical properties of polymer, hybrid, and all-inorganic porous coatings was performed using nanoindentation hardness measurements (Figure 6). Our results reveal that swelling (Figure 6a,b) as well as the infiltration of the swollen polymers with TMA and DEZ (Figure 6c,d) does alter the mechanical properties of the polymers. However, porous AlO_x_ and ZnO coatings obtained as a result of polymer removal demonstrate a substantial increase in hardness (Figure 6c,d). Note, that porous AlO_x_ and ZnO structures obtained with the PIM-1 template (Figure 6c,d) are significantly harder than those obtained with PS75-b-P4VP25 BCP templates. This is attributed to the denser morphology of the PIM-designed coatings. However, compared to the porous ZnO coatings obtained using PS75-b-P4VP25 BCP templates, the porous AlO_x_ fabricated with the BCP template is much softer, possibly due to the morphology and nature of porosity in the coatings, which can be seen in the SEM images above (Figure 5b,d). Moreover, at the polymer removal temperature (~450 °C), the porous AlO_x_ remains amorphous [31,32,33], whereas the temperature was sufficient to crystallize ZnO coating [8,34,35]. Though grain refinement can help to increase the yield and tensile strength of materials leading to the higher mechanical robustness of amorphous structures in comparison to the crystalline structures, this is usually true for bulk materials only. With the porosity introduction, amorphous materials experience a reduction in the resistance to the propagation of dislocations and, thus, are much more prone to damage [18,36,37].

In order to test the effect of crystallinity on the hardness, we annealed the coatings at 900 °C for 1 h (Figure 7). Crystallization of AlO_x_ was reported to occur at above 800 °C [31,38], leading to the formation of the θ-Al_2_O_3_ and α-Al_2_O_3_ phases. However, ZnO has been reported to crystallize at a much lower temperature of 300 °C [35]. The crystalline nature of the ZnO conformal coatings synthesized using polymer templates was confirmed in our previous studies [7,8].

The annealing of all ZnO and AlO_x_ coatings except for AlO_x_ coatings obtained using the PS75-b-P4VP25 BCP template increased the hardness, possibly due to the porosity and defects reduction as evident in the AFM images of the coatings (Figure 8). Note, the hardness of porous ZnO obtained using the PS75-b-P4VP25 BCP template (Figure 7d) was significantly enhanced with annealing from 1.7 GPa to 6.5 GPa at a depth of 45 nm. The increase in hardness values agrees with the previously reported annealing effect for porous anodic aluminum oxide coatings [39]. However, for AlO_x_ coatings obtained using PS75-b-P4VP25 BCP template, at an indenter depth below 85 nm for a ~1 µm thick coating, the hardness of the porous AlO_x_ (BCP) (Figure 7b) remained the same even after annealing. The improvements in crystallinity can potentially lead to higher values of hardness since the crystals can exhibit exceptional hardness in specific crystallographic directions. However, polycrystalline materials consisting of randomly oriented crystalline domains exhibit the averaging effect on their hardness properties [36,37]. In the case of porous materials, the large volume of voids can diminish the mechanical properties of the materials regardless of their crystallinity [40]. However, previously, a substantial increase in the hardness was observed for highly porous Cu and CuNi structures synthesized via infiltration of the patterned three-dimensional polymer templates with inorganic salts followed by polymer removal and reductive treatment [41]. Therefore, it is difficult to draw unambiguous conclusions about the trends accompanying the structural transformation in highly porous materials, and further investigations are needed. However, it is worth noting that we previously reported high thermal stability of the catalytic AlO_x_-based structure synthesized using the removal of PS75-b-P4VP25 BCP support [23]. Moreover, we did not observe any noticeable effect of temperature of annealing on the catalytic performance. These data agree with the unchanged mechanical properties of AlO_x_-based structures synthesized using BCP templates.

Since QCM measurements cannot be performed on the annealed samples because of the instability of the QCM crystals, we studied the change in the porosities of the synthesized and annealed porous AlO_x_ and ZnO coatings by analysis of their refractive indices and percentage porosity (Figure 8). Porous AlO_x_ obtained using PIM-1 templates did not reveal any changes in the refractive indices over the broad spectral range (Figure 8a). AFM data obtained for such samples before and after annealing indicate nearly no changes in the morphology of the coatings, which is consistent with no change in refractive index and porosity (~20%) of the coating obtained from the ellipsometry measurement. In turn, we found that the refractive index increases slightly with annealing at 900 °C for AlO_x_ obtained with BCP (Figure 8b) and ZnO coatings with BCP and PIM-1 templates, respectively (Figure 8c,d). The largest increase in refractive index, from 1.15 to 1.5, is observed for the ZnO coating synthesized with the PS75-b-P4VP25 BCP template (Figure 8d). This observation is attributed to the crystallinity of the coating resulting from grain growth and a decrease in porosity as well as densification of the coatings. Previously, it was reported that an increase in annealing temperature resulted in a shift in the dispersive curve of refractive index to higher values [42,43], which is in agreement with the previously reported tendency of grains to coalesce and form bigger grains [43]. These porosity estimations are also in agreement with the observations from the AFM images and refractive index data. After annealing at 900 °C, we observe a minimal decrease in porosity, from ~40% to ~30% for the ZnO (PIM) coating (Figure 8c) and from ~80% to ~73% for the AlO_x_ (BCP) coating (Figure 8b). Meanwhile, annealing of the ZnO (BCP) coating resulted in a substantial decrease in porosity, from ~74.3% to ~43.6% (Figure 8d). This further suggests that the films retain some level of porosity after annealing. The AFM images for the porous ZnO coatings (Figure 8c,d) show evenly distributed spherical ZnO nanoparticles prior to annealing at 900 °C. After annealing, grain growth and coalescence are observed. As for the porous AlO_x_ coating synthesized with the BCP template (Figure 8b), porous AlO_x_ networks of interconnected nanotubes are evident.

We also tested the effect of the water freezing inside the pores on the mechanical stability of the pores in the synthesized AlO_x_ and ZnO porous structures (Figure 9a). Therefore, we compared the refractive indices of our coatings before freezing in water, and after their defrosting (Figure 9). We placed the porous coatings in a Petri dish filled with water enough to submerge the samples in before placing them in the freezer for about 12 h. After freezing, the samples were defrosted on a hot plate at 60 °C for 2 h to ensure their drying. As in the case of annealing, the freezing of the water in the pores of AlO_x_ obtained using PIM-1 templates does not cause any change in its refractive index. The rest of the samples reveal a minor change in the refractive indices (Figure 9b–e). Therefore, we conclude that porous AlO_x_ and ZnO are synthesized using polymer templates such as mechanically stable and can be utilized in freezing weather conditions without compromising their optical properties [9,10].

## 4. Conclusions

Here, we followed the porosity evolution and pores’ accessibility during different stages of the SIS-assisted synthesis of nanoporous aluminum oxide and zinc oxide nanoporous conformal coatings such as polymer template swelling, infiltration of the polymer with alumina and zinc oxide precursors, and polymer removal. Using the QCM technique, we analyzed the room temperature accessibility of pores to water and ethanol in porous polymer templates using polystyrene-block-poly-4-vinyl pyridine (PS75-b-P4VP25) and polymer of intrinsic microporosity (PIM-1), polymer templates modified by swelling, and porous inorganic coatings such as AlO_x_ and ZnO synthesized by SIS using such templates. Our results indicate a higher wettability of AlO_x_ than of ZnO nanoporous coatings for water and ethanol. Using analysis of data obtained in nanoindentation tests, we demonstrated that the nature of the polymer templates plays an important role in determining the mechanical characteristics of the metal oxide films, with PIM-based coatings showing improved hardness over the BCP-based coatings. QCM and optical reflectance measurements were used to assess the stability of materials upon exposure to extreme temperature conditions. Results indicated that annealing of the coatings at 900 °C allowed them to further improve their mechanical characteristics though the overall structural integrity remained unchanged. All the films demonstrated stability upon freezing experiments suggesting that though the porosity is highly accessible to solvents, solvent freezing does not damage the structures.

## Figures and Tables

**Figure 1 polymers-15-04088-f001:**
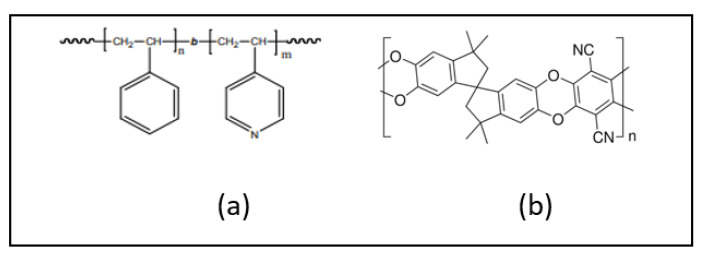
Chemical structures of polymers used. (**a**) Polystyrene-block-poly-4-vinyl pyridine (PS75-b-P4VP25) and (**b**) polymer of intrinsic microporosity (PIM-1).

**Figure 2 polymers-15-04088-f002:**
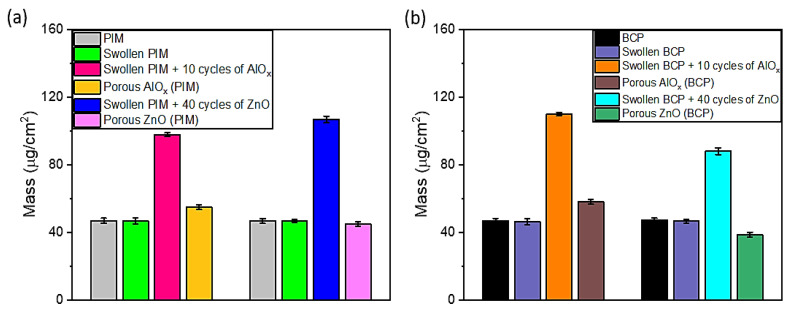
Summary of the mass changes measured by the QCM for the initial polymers, after swelling, SIS of polymers with Alumina and ZnO, and polymer removal (all changes in frequency are normalized by the frequency of a clean QCM): (**a**) for PIM-based coatings and (**b**) BCP-based coatings.

**Figure 3 polymers-15-04088-f003:**
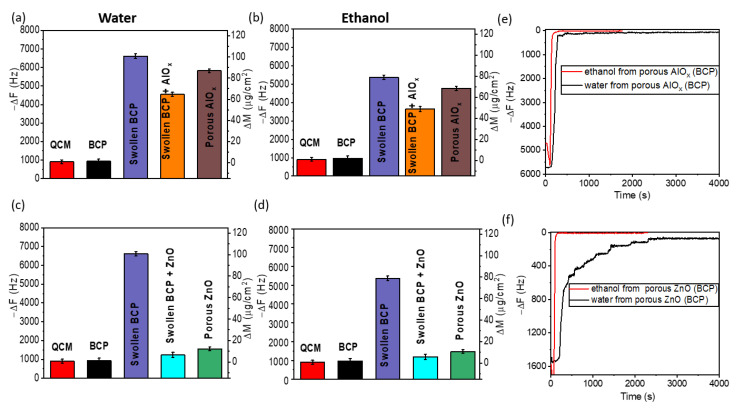
QCM analysis of the porosity for the non-swollen BCP, swollen BCP, SIS with AlO_x_ and ZnO, porous AlO_x_ and porous ZnO after polymer removal in water and ethanol (**a**–**d**): Absolute value of frequency with a corresponding mass of solvent penetration in water (**a**,**c**) and ethanol (**b**,**d**). The plot of frequency vs. time analyzing the drying curve for (**e**) porous alumina and (**f**) porous ZnO coatings after removal in ethanol and water.

**Figure 4 polymers-15-04088-f004:**
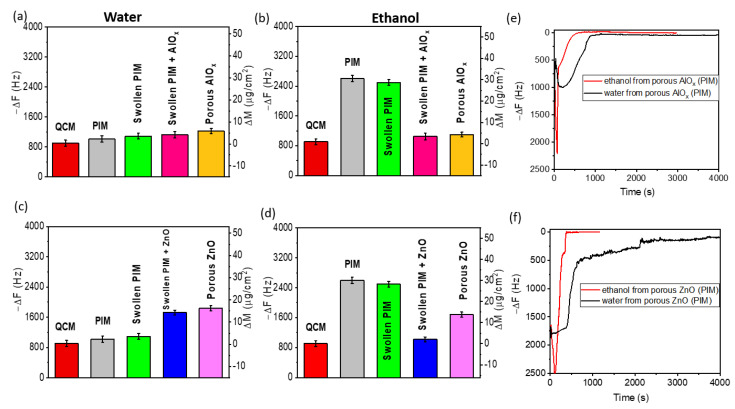
QCM analysis of the porosity for the non-swollen PIM, swollen PIM, SIS with AlOx and ZnO, porous AlOx, and porous ZnO after polymer removal in water and ethanol (**a**–**d**): Absolute value of frequency with a corresponding mass of solvent penetration in water (**a**,**c**) and ethanol (**b**,**d**). The plot of frequency vs. time analyzing the drying curve for (**e**) porous AlOx and (**f**) porous ZnO coatings after removal in ethanol and water.

**Figure 5 polymers-15-04088-f005:**
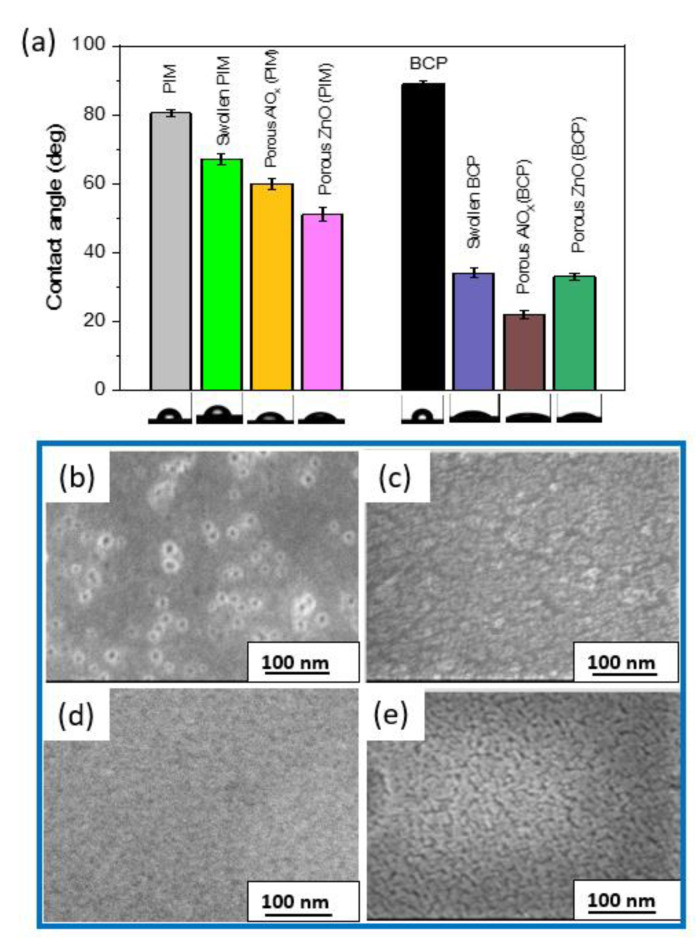
Summary of wetting properties based on the contact angle measurements (**a**) and SEM images of porous (**b**) AlO_x_ (BCP), (**c**) porous ZnO (BCP), (**d**) porous AlO_x_ (PIM), (**e**) porous ZnO (PIM) coating.

**Figure 6 polymers-15-04088-f006:**
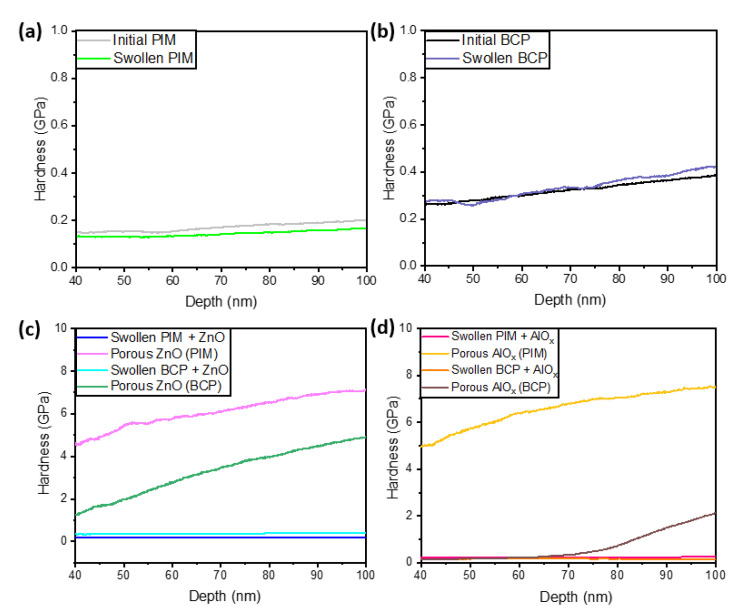
Mechanical properties of the non-swelled polymers (PIM, BCP), swelled polymers, infiltrated polymers, and porous inorganic coatings after polymer removal (**a**–**d**): (**a**) non-swelled PIM vs. swelled PIM, (**b**) initial BCP vs. swelled BCP, (**c**) SIS of PIM with ZnO, SIS of BCP with ZnO, porous ZnO via SIS of PIM, porous ZnO via SIS of BCP, (**d**) SIS of PIM with AlO_x_, SIS of BCP with AlO_x_, porous AlO_x_ via SIS of PIM, and porous AlO_x_ via SIS of BCP.

**Figure 7 polymers-15-04088-f007:**
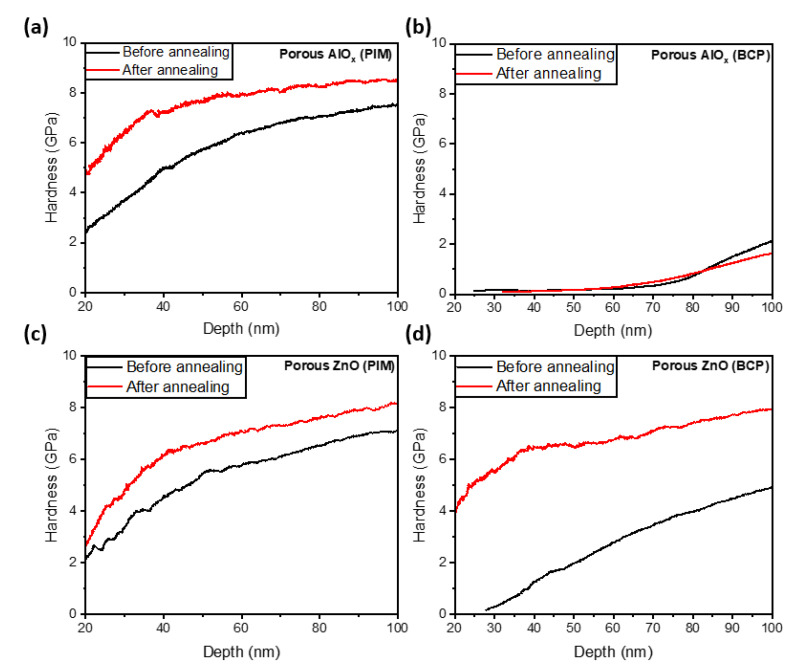
Mechanical properties of porous AlO_x_ via SIS of PIM-1 and BCP ((**a**,**b**), respectively) and porous ZnO films via SIS of PIM-1 and BCP templates ((**c**,**d**), respectively) before and after annealing at 900 °C for 1 h.

**Figure 8 polymers-15-04088-f008:**
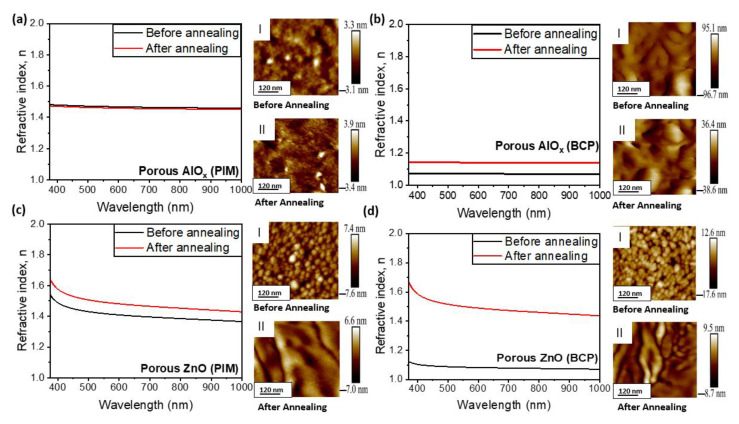
Refractive indices and AFM images of porous AlO_x_ obtained via SIS of PIM-1 and PS75-b-P4VP25 BCP templates ((**a**,**b**), respectively) and porous ZnO films via SIS of PIM and BCP films (**c**,**d**) before annealing (I) and after annealing (II) at 900 °C for 1 h.

**Figure 9 polymers-15-04088-f009:**
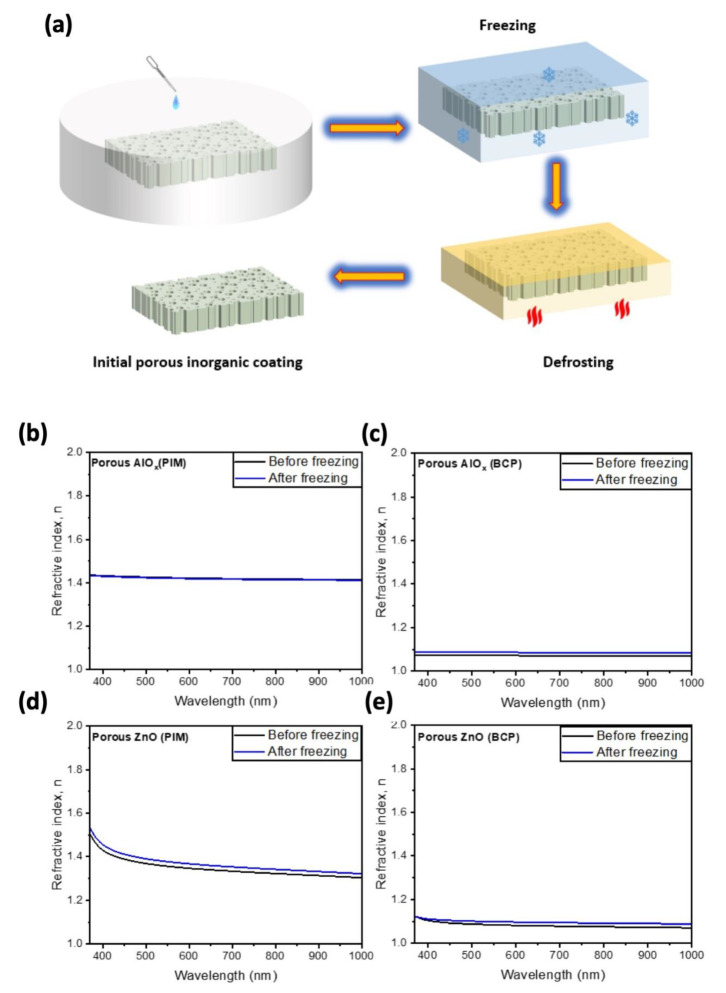
(**a**) Schematics depicting the experimental setup for investigation of the mechanical stability of pores in harsh weather conditions. Refractive indices of porous AlO_x_ via SIS of PIM and BCP (**b**,**c**) and porous ZnO films via SIS of PIM and BCP films (**d**,**e**) before freezing and after freezing.

**Table 1 polymers-15-04088-t001:** List all the samples used in the study and the corresponding color code used for each of them.

 Base QCM crystal	
 Initial PIM template	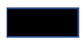 Initial BCP template
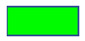 Swollen PIM	 Swollen BCP
 Swollen PIM + 10 cycles of AlO_x_ SIS	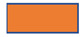 Swollen BCP + 10 cycles of AlO_x_ SIS
 Porous AlO_x_ after PIM removal	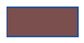 Porous AlO_x_ after BCP removal
 Swollen PIM + 40 cycles of ZnO SIS	 Swollen BCP + 40 cycles of ZnO SIS
 Porous ZnO after PIM removal	 Porous ZnO after BCP removal

**Table 2 polymers-15-04088-t002:** Summary of the effective pore volume in the non-swollen polymers and the swollen polymers accessible to water and ethanol at room temperature when neglecting the decoupling of the polymer due to its visco-elastic nature. The effective pore volume values were calculated using Equation (3).

Sample	Volume of Pores Filled with Water (cm^3^)	Volume of Pores Filled with Ethanol (cm^3^)
Non- Swollen PS75-P4VP25	-	-
Swollen PS75-P4VP25	5.26 ± 0.23 × 10^−4^	5.44 ± 0.22 × 10^−4^
Non- Swollen PIM-1	2.69 ± 0.13 × 10^−5^	2.35 ± 0.13 × 10^−4^
Swollen PIM-1	3.43 ± 0.15 × 10^−5^	2.23 ± 0.14 × 10^−4^

**Table 3 polymers-15-04088-t003:** Summary of the effective pore volume in the infiltrated polymer coatings and porous inorganic coatings accessible to water and ethanol at room temperature when neglecting the decoupling of the polymer due to visco-elastic nature. The effective pore volume values were calculated using Equation (3).

Sample	Volume of Pores Filled with Water (cm^3^)	Volume of Pores Filled with Ethanol (cm^3^)
Swollen PS75-P4VP25 + AlO_x_	3.44 ± 0.08 × 10^−4^	3.53 ± 0.12 × 10^−4^
Swollen PS75-P4VP25 + ZnO	4.60 ± 0.15 × 10^−5^	7.57 ± 0.23 × 10^−5^
AlO_x_-PS75-P4VP25 (after UV treatment)	4.58 ± 0.11 × 10^−4^	4.80 ± 0.14 × 10^−4^
ZnO-PS75-P4VP25 (after UV treatment)	7.44 ± 0.26 × 10^−5^	1.07 ± 0.16 × 10^−4^
Swollen PIM-1 + AlOx	3.73 ± 0.15 × 10^−5^	6.74 ± 0.13 × 10^−5^
Swollen PIM-1 + ZnO	9.04 ± 0.16 × 10^−5^	5.51 ± 0.23 × 10^−5^
AlO_x_-PIM-1 (after UV treatment)	4.61 ± 0.14 × 10^−5^	5.40 ± 0.15 × 10^−5^
ZnO-PIM-1 (after UV treatment)	1.02 ± 0.07 × 10^−4^	1.30 ± 0.06 × 10^−4^

## Data Availability

The authors confirm that the data supporting the findings of this study are available within the article.

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
