# Peer review of "Accessibility and Mechanical Stability of Nanoporous Zinc Oxide and Aluminum Oxide Coatings Synthesized via Infiltration of Polymer Templates"

_polymers, 2023, doi:10.3390/polym15204088_

Round 1

Reviewer 1 Report

The authors prepared the nanoporous structures by SIS method and evaluated the effects of the polymer types on the several properties. The authors demonstrate enough data to support the conclusion in the manuscript.  I think that the manuscript can be published after minor revision. Comments are below.

1. In page 8 line 295, the authors say "due to the opening of the micelles after swelling". What is the meaning of "opening"? 

2. In Figure 8, the authors tested the effect of the water freezing. What is the meaning of this experiment?  How is water freezing important for nanoporous structures ?  If possible, it is better to include the motivation for performing the water freezing experiment in the manuscript.

Author Response

We would like to thank the editor and reviewers for their time, careful examination of our manuscript, and useful comments. We hope that you will find the revised version of our manuscript suitable for publication in Polymers. Our detailed response to the comments is given below.

Reviewer 1:

The authors prepared the nanoporous structures by SIS method and evaluated the effects of the polymer types on the several properties. The authors demonstrate enough data to support the conclusion in the manuscript.  I think that the manuscript can be published after minor revision. Comments are below.

Response:  We thank the reviewer for the positive feedback on our study.

  1. In page 8 line 295, the authors say "due to the opening of the micelles after swelling". What is the meaning of "opening"?

Response: We appreciate the reviewer’s comment. By opening the micelle, we mean that the P4VP domain of the block copolymer expands in response to swelling in ethanol, thereby creating an opening for penetration. We clarified this point in the revised version of the manuscript.

  1. In Figure 8, the authors tested the effect of the water freezing. What is the meaning of this experiment?  How is water freezing important for nanoporous structures ?  If possible, it is better to include the motivation for performing the water freezing experiment in the manuscript.

Response: We thank the reviewer for the valuable insight. The porous inorganic coatings can be employed in various environmental conditions depending on the application requirements, which may potentially lead to the exposure of materials to high and low temperatures in the presence of humidity. For example, SIS-designed nanoporous alumina films are considered for antireflective coatings applied on lenses, windows, and solar panels. Therefore, considering the high degree of porosity accessibility to the solvents, it is important to demonstrate the ability of our materials to withstand freezing conditions without compromising their properties. We expanded the motivation for this experiment in lines in the introduction section.

Reviewer 2 Report

The presented manuscript is dedicated to preparation of porous coatings of zinc oxide and aluminium oxide by the use of sequential infiltration synthesis method. It was shown that the properties of the final coatings depend on the reaction conditions and moreover on the nature of the used polymeric precursor. However there are some notices:

- it is not clear from the experimental part about swelling of polymeric coatings. Were they dried after swelling or they contain ethanol? If they were dried, please provide drying conditions (time, temperture, vacuum etc), because it may significantly influence on the coating morphology. If they were not dried, it is not quite correct to analyze such coatings by water contact angle.

- it would be better to provide chemical structures of the used polymers and explain in more details the choice of such different by nature polymers. Especially, explain choice of block-copolymer with exactly such sctructure.

- on page 10 authors speculate that the mechanical properties of ZnO higher because of its crystallinity. However there is no data like XPS, which definitely could confirm it.

- it should be explained why for coatings, made from PIM the mechanical properties are similar in comparison with BCP.

- page 8. Authors claim that swollen PIM coating is hydrophobic, but is is clearly seen, that contact angle is about 65, which is hydrophilic. 

Round 2

Reviewer 2 Report

Authors have improved the manuscript and now it can be recommended for publication